# Fusing Temporal Graphs into Transformers for Time-Sensitive Question Answering

**Xin Su**[1,2]    **Phillip Howard**[2]    **Nagib Hakim**[2]    **Steven Bethard**[1]
[1]University of Arizona    [2]Intel Labs
{xinsu, bethard}@arizona.edu, {phillip.r.howard, nagib.hakim}@intel.com

## Abstract

Answering time-sensitive questions from long documents requires temporal reasoning over the times in questions and documents. An important open question is whether large language models can perform such reasoning solely using a provided text document, or whether they can benefit from additional temporal information extracted using other systems. We address this research question by applying existing temporal information extraction systems to construct temporal graphs of events, times, and temporal relations in questions and documents. We then investigate different approaches for fusing these graphs into Transformer models. Experimental results show that our proposed approach for fusing temporal graphs into input text substantially enhances the temporal reasoning capabilities of Transformer models with or without fine-tuning. Additionally, our proposed method outperforms various graph convolution-based approaches and establishes a new state-of-the-art performance on SituatedQA and three splits of TimeQA.

## 1 Introduction

Long-document time-sensitive question answering (Chen et al., 2021) requires temporal reasoning over the events and times in a question and an accompanying long context document. Answering such questions is a challenging task in natural language processing (NLP) as models must comprehend and interpret the temporal scope of the question as well as the associated temporal information dispersed throughout the long document. For example, consider the time-sensitive questions about George Washington's position provided in Figure 1. The relevant events and temporal information regarding George Washington's position are scattered across many different sentences in the context document. Since there is no one single text segment containing the answer, the model must integrate and reason over events and times

| Context Document |
| --- |
| George Washington (February 22, 1732 – December 14, 1799) was an American political leader. ... TLDR (hundred words) |
| Congress created the Continental Army on June 14, 1775, and Samuel and John Adams nominated Washington to become its Commander-in-Chief. ... TLDR (hundred words) |
| Washington bade farewell to his officers at Fraunces Tavern in December 1783 and resigned his commission days later. In 1788, the Board of Visitors of the College of William & Mary decided to re-establish the position of Chancellor, and elected Washington to the office on January 18. ... TLDR (hundred words) |
| He started as the president of United Sates in Jan 1788, …, in 1798, one year after that, he stepped down his presidency position. |

**Q1:** What was George Washington's position between 1776 - 1780? **A:** Commander-in-Chief

**Q2:** What was George Washington's position from 1790 - 1797? **A:** Presidency, Chancellor

Figure 1: Time-sensitive question-answer pairs with a context document. The times and answers are in red and blue, respectively.

throughout the context document. Additionally, this example illustrates how changing the time expression in the question may also result in a change in the answer: in this case, replacing *between 1776 - 1780* with *from 1790 to 1797* changes the answer from *Commander-in-Chief* to *Presidency* and *Chancellor*.

Though not designed directly for question answering, there is a substantial amount of research on temporal information extraction (Chambers et al., 2014; Ning et al., 2018a; Zhang and Xue, 2018, 2019; Han et al., 2019; Ning et al., 2019; Vashishtha et al., 2019; Ballesteros et al., 2020; Yao et al., 2020; Zhang et al., 2022a). Such models can help reveal the structure of the timeline underlying a document. However, there is little existing research on combining such information extraction systems with question answering Transformer models (Izacard and Grave, 2021) to effectively reason over temporal information in long documents.

In this work, we utilize existing temporal information extraction systems to construct temporal

graphs and investigate different fusion methods to inject them into Transformer models.

We evaluate the effectiveness of each temporal graph fusion approach on long-document time-sensitive question answering datasets. Our contributions are as follows:

1. We introduce a simple but novel approach to fuse temporal graphs into the input text of question answering Transformer models.

2. We compare our method with prior approaches such as fusion via graph convolutions, and show that our input fusion method outperforms these alternative approaches.

3. We demonstrate that our input fusion approach can be used seamlessly with large language models in an in-context learning setting.

4. We perform a detailed error analysis, revealing the efficacy of our method in fixing temporal reasoning errors in Transformer models.

## 2   Related Work

### 2.1   Extracting Temporal Graphs

Research on extracting temporal graphs from text can be grouped into the extraction of event and time graphs (Chambers et al., 2014; Ning et al., 2018b), contextualized event graphs (Madaan and Yang, 2021), and temporal dependency trees and graphs (Zhang and Xue, 2018, 2019; Yao et al., 2020; Ross et al., 2020; Choubey and Huang, 2022; Mathur et al., 2022). Additionally, some prior work has focused on the problem of extracting temporal relations between times and events (Ning et al., 2018a, 2019; Vashishtha et al., 2019; Han et al., 2019; Ballesteros et al., 2020; Zhang et al., 2022a). The outputs of these temporal relation extraction systems are often used to construct temporal graphs. In this work, we use CAEVO (Chambers et al., 2014) and SUTime (Chang and Manning, 2012) to construct our temporal graphs because they are publicly available (unlike more recent models such as the temporal dependency graph parser proposed by Mathur et al. (2022)) and can easily scale to large amounts of long documents without requiring additional training.

### 2.2   Temporal Question Answering

Jia et al. (2018) decomposes questions and applies temporal constraints to allow general question-answering systems to answer knowledge-base-temporal questions. Saxena et al. (2021), Jia et al. (2021), Mavromatis et al. (2021), and Sharma et al. (2023) use time-aware embeddings to reason over temporal knowledge graphs. Similar to our work, Huang et al. (2022) and Shang et al. (2021) answer temporal questions on text, but they focus on temporal event ordering questions over short texts rather than time-sensitive questions over long documents. Li et al. (2023) focus on exploring large language models for information extraction in structured temporal contexts. They represent the extracted time-sensitive information in code and then execute Python scripts to derive the answers. In contrast, we concentrate on temporal reasoning in the reading comprehension setting, using unstructured long documents to deduce answers. This poses more challenges in information extraction and involves more complex reasoning, which motivates our integration of existing temporal information extraction systems with transformer-based language models. The most similar work to ours is Mathur et al. (2022), which extracts temporal dependency graphs and merges them with Transformer models using learnable attention mask weights. We compare directly to this approach, and also explore both graph convolutions and input modifications as alternatives to fusing temporal graphs into Transformer models.

### 2.3   Fusing Graphs into Transformer Models

The most common approaches for fusing graphs into Transformer models are graph neural networks (GNN) and self-attention. In the GNN-based approach, a GNN is used to encode and learn graph representations which are then fused into the Transformer model (Yang et al., 2019; Feng et al., 2020; Yasunaga et al., 2021; Zhang et al., 2022b). In the self-attention approach, the relations in the graphs are converted into token-to-token relations and are then fused into the self-attention mechanism. For example, Wang et al. (2020) uses relative position encoding (Shaw et al., 2018) to encode a database schema graph into the BERT representation. Similarly, Bai et al. (2021) utilize attention masks to fuse syntax trees into Transformer models. We explore GNN-based fusion of temporal graphs into question answering models, comparing this approach to the attention-based approach of Mathur et al. (2022), as well as our simpler approach which fuses the temporal graph directly into the Transformer model's input.

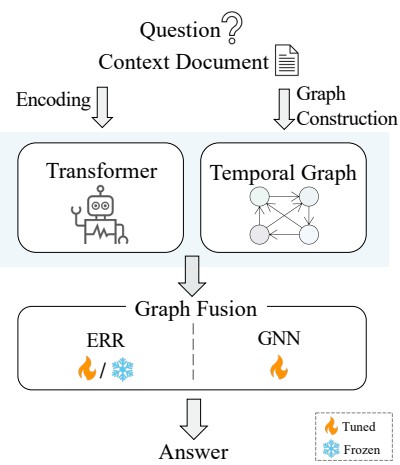

Figure 2: Overview of our approach. The ERR method allows for optional fine-tuning of the Transformer model, whereas for the GNN method requires fine-tuning.

# 3 Method

Our approach applies temporal information extraction systems to construct temporal graphs and then fuses the graphs into pre-trained Transformer models. We consider two fusion methods:

1. Explicit edge representation fusion (ERR): a simple but novel approach to fuse the graphs into the input text.

2. Graph neural network fusion: a GNN is used to fuse the graphs into the token embeddings or the last hidden layer representations (i.e., contextualized embeddings) of the Transformer model.

The overall approach is illustrated in Figure 2.

## 3.1 Graph Construction

Given a time-sensitive question and a corresponding context document, we construct a directed temporal graph where events and time expressions are nodes and temporal relations are edges of the type BEFORE, AFTER, INCLUDES, INCLUDED BY, SIMULTANEOUS, or OVERLAP.

We extract the single timestamp included in each question, which is either explicitly provided by the dataset (as in SituatedQA (Zhang and Choi, 2021)) or alternatively is extracted via simple regular expressions (the regular expressions we use achieve 100% extraction accuracy on TimeQA (Chen et al., 2021)). We add a single question-time node to the graph for this time.

For the document, we apply CAEVO[1] to iden-

---

[1] https://github.com/nchambers/caevo

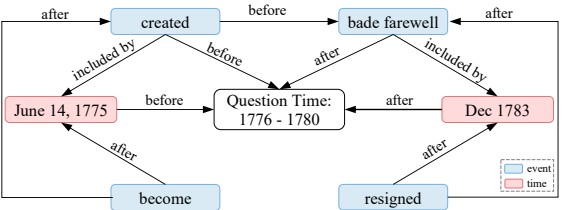

Figure 3: Temporal graph example.

tify the events, time expressions, and the temporal relations between them. CAEVO follows the standard convention in temporal information extraction that events are only simple actions (typically verbs) with linking of these actions to subjects, objects, and other arguments left to dependency parsers (Pustejovsky et al., 2003; Verhagen et al., 2009, 2010; UzZaman et al., 2013). We add document-event and document-time nodes for each identified event and time, respectively, and add edges between nodes for each identified temporal relation.

To link the question-time node to the document-time nodes, we use SUTime[2] to normalize time expressions to time intervals, and deterministically compute temporal relations between question time and document times as edges. For example, given a document-time node "the year 2022" and a question-time node "from 1789 to 1797" from the question "What was George Washington's position from 1789 to 1797?", the times will be normalized to [2022-01-01, 2022-12-31] and [1789-01-01, 1797-12-31] respectively, and the temporal relation between them can then be computed as AFTER.

To link the question-time node to document events, for each document-event node, we calculate the shortest path in the temporal graph between it and the question-time node and recursively apply standard transitivity rules (see Appendix A.1) along the path to infer the temporal relation. For example, given a path A is BEFORE B and B INCLUDES C, we can infer the relation between A and C is BEFORE. An example of a constructed temporal graph for Q1 in Figure 1 is illustrated in Figure 3.

## 3.2 Graph Fusion

For both fusion methods, we concatenate the question and corresponding context document as an input sequence to the Transformer model. For example, given the question and document from Figure 1 Q1, the input is:

---

[2] https://nlp.stanford.edu/software/sutime.html

*question: What was George Washington's position between 1776 - 1780? context: ...Congress created the Continental Army on June 14, 1775 ...*

### 3.2.1 Explicit Edge Representation

In the ERR method, we mark a temporal graph's nodes and edges in the input sequence, using `<question time>` and `</question time>` to mark the question-time node and relation markers such as `<before>` and `</before>` to mark the nodes in the context document and their relations to the question time. Thus, the ERR input for the above example is:

*question: What was George Washington's position <question time>between 1776-1780</question time>? context: ...Congress <before>created</before> the Continental Army on <before>June 14, 1775</before>...*

This approach aims to make the model learn to attend to parts of the input sequence that may contain answer information. For instance, the model may learn that information related to the answer may be found near markers such as `<overlap>`, `<includes>`, `<included by>`, and `<simultaneous>`. Additionally, the model may learn that answer-related information may exist between `<before>` and `<after>`, even if the answer does not have any nearby temporal information.

### 3.2.2 GNN-based Fusion

In GNN-based fusion, we add `<e>` and `</e>` markers around each node, and apply a relational graph convolution (RelGraphConv; Schlichtkrull et al., 2018) over the marked nodes. RelGraphConv is a variant of graph convolution (GCN; Kipf and Welling, 2017) that can learn different transformations for different relation types. We employ the RelGraphConv to encode a temporal graph and update the Transformer encoder's token embedding layer or last hidden layer representations (i.e., contextualized embeddings). We utilize the RelGraphConv in its original form without any modifications.

Formally, given a temporal graph $G = (V, E)$, we use representations of the `<e>` markers from the Transformer model's token embedding layer or the last hidden layer as initial node embeddings. The output of layer $l + 1$ for node $i \in V$ is:

$$h_i^{l+1} = \sigma(\sum_{r \in R} \sum_{j \in \mathcal{N}^r(i)} \frac{1}{c_{i,r}} W_r^{(l)} h_j^{(l)} + W_0^{(l)} h_i^{(l)})$$

where $\mathcal{N}^r(i)$ denotes all neighbor nodes that have relation $r$ with node $i$, $\frac{1}{c_{i,r}}$ is a normalization constant that can be learned or manually specified, $\sigma$ is the activation function, $W_0$ is the self-loop weight, and $W_r$ is the learnable weights. We refer readers to Schlichtkrull et al. (2018) for more details.

## 4 Experiments

### 4.1 Datasets

We evaluate on two time-sensitive question answering datasets: Time-Sensitive Question Answering (TimeQA; Chen et al., 2021) and SituatedQA (Zhang and Choi, 2021). We briefly describe these two datasets below and provide statistics on each dataset in Table 9 of Appendix A.6.

**TimeQA** The TimeQA dataset is comprised of time-sensitive questions about time-evolving facts paired with long Wikipedia pages as context. The dataset has two non-overlapping splits generated by diverse templates which are referred to as TimeQA Easy and TimeQA Hard, with each split containing 20k question-answer pairs regarding 5.5K time-evolving facts and 70 relations. TimeQA Easy contains questions which typically include time expressions that are explicitly mentioned in the context document. In contrast, TimeQA Hard has time expressions that are not explicitly specified and therefore require more advanced temporal reasoning. For example, both questions in Figure 1 are hard questions, but if we replace the time expressions in the questions with *In 1788*, they will become easy questions. In addition, smaller Human-paraphrased Easy and Hard splits are also provided.

**SituatedQA** SituatedQA is an open-domain question answering dataset comprising two subsets: Temporal SituatedQA and Geographical SituatedQA. We focus on Temporal SituatedQA, which we will hereafter refer to as SituatedQA. Each question in SituatedQA is accompanied by a temporal annotation that could change the answer to the question if it is modified. For instance, the question "Which COVID-19 vaccines have been authorized for adults in the US as of Jan 2021?" has a corresponding answer of "Moderna, Pfizer." However, if we change the time to "Apr 10, 2021," the answer becomes "Moderna, Pfizer, J&J." As SituatedQA is a re-annotation of a subset of NQ-Open (Kwiatkowski et al., 2019a; Lee et al., 2019), we use the top 100 passages retrieved from Wikipedia

| Method | Model Size | External Tool (Public Avail.) | Add. Data | Knowl. Fusion |
|---|---|---|---|---|
| BigBird (Chen et al., 2021) | 128M | - | NaturalQ | - |
| FiD & Replicated FiD (Chen et al., 2021) | 220M | - | NaturalQ | - |
| DocTime (Mathur et al., 2022) | 220M | CAEVO (✓), TDGP (✗) | NaturalQ | Attention |
| BigBird + MTL (Chen et al., 2023a) | 128M | TT (✓), TR (✗) | TriviaQA | Multi-Task |
| Longformer + MTL (Chen et al., 2023a) | 435M | TT (✓), TR (✗) | TriviaQA | Multi-Task |
| DPR + Query Modified (Zhang and Choi, 2021) | 110 M | - | - | Question Text |
| TSM (Cole et al., 2023) | 11B | SUTime (✓) | Entire Wiki | Pre-Training |
| LongT5$_{ERR}$ (Ours) | 250M | SUTime (✓) | NaturalQ | Input Text |

Table 1: Comparison of methods, base models' size, external tools, additional data, and temporal knowledge fusion methods. NaturalQ: Natural Questions, TDGP: Temporal Dependency Graph Parser, TT: BERT-based Temporal Tagger, TR: timestamps retriever.

by FiD (Izacard and Grave, 2021)[3] for each question as context documents.

**Evaluation Metrics**  We use the official evaluation methods and metrics provided in the code release of the datasets. For TimeQA, we report the exact match and F1 scores. For SituatedQA, we report the exact match score.

### 4.2  Baselines

We compare our models with the previous state-of-the-art on the TimeQA and SituatedQA, which we describe in this section and summarize in Table 1.

For TimeQA, we compare against:

**FiD & BigBird**  Chen et al. (2021) adapt two long-document question answering models, BigBird (Zaheer et al., 2020) and FiD (Izacard and Grave, 2021), to the TimeQA dataset. Before fine-tuning the models on TimeQA, they also fine-tune on Natural Questions (Kwiatkowski et al., 2019b) and TriviaQA (Joshi et al., 2017), finding that training on Natural Questions results in the best performance. The best model from Chen et al. (2021) on TimeQA Easy and Hard is FiD, while BigBird performs best on the Human-paraphrased TimeQA Easy and Hard splits.

**Replicated FiD**  We also report our replication of Chen et al. (2021)'s FiD model using the code provided by their GitHub repository[4], but even with extensive hyperparameter tuning, we were unable to reproduce their reported performance of the FiD model on TimeQA Easy[5].

**DocTime**  DocTime (Mathur et al., 2022) first uses CAEVO to identify events and time expressions in context documents. Then, a custom-trained temporal dependency parser is applied to parse temporal dependency graphs. Finally, the parsed temporal dependency graphs are fused into the attention mechanism of the FiD model, which is fine-tuned on Natural Questions.

**BigBird + MTL & Longformer + MTL**  Chen et al. (2023a) inject temporal information into long text question answering systems using a multi-task learning (MTL) approach. They train BigBird and Longformer models on time-sensitive question answering tasks, along with three auxiliary temporal-awareness tasks. They explore first fine-tuning on Natural Questions and TriviaQA datasets, finding that TriviaQA results in the best performance. BigBird + MTL is their best model on TimeQA Easy, while Longformer + MTL performs best on TimeQA Hard.

For SituatedQA, we compare against:

**DPR + Query Modified**  Zhang and Choi (2021) adapt a retrieval-based model, DPR (Karpukhin et al., 2020), for SituatedQA. They first fine-tune the retriever and reader of DPR on Natural Questions and then further fine-tune on SituatedQA.

**TSM** (Cole et al., 2023) uses SuTime to identify and mask time expressions throughout all Wikipedia documents. They then fine-tune T5-1.1-XXL (Raffel et al., 2020) to predict the masked time expressions. Finally, they fine-tune the resulting T5-1.1-XXL model on SituatedQA.

### 4.3  Implementation Details

We use LongT5-base (Guo et al., 2022), a transformer sequence-to-sequence model, as our base model. Our experiments demonstrate that LongT5 outperforms the FiD model (Izacard and Grave, 2020) commonly used on long-document question answering tasks. To fairly compare with previous work (Chen et al., 2021; Mathur et al., 2022), we pre-train the LongT5 model on Natural Questions (Kwiatkowski et al., 2019a) and then fine-tune it

---

[3]https://github.com/facebookresearch/FiD
[4]wenhuchen/Time-Sensitive-QA
[5]Other researchers have reported the same issue on GitHub

| | TimeQA Easy | | TimeQA Hard | | TimeQA$_{\text{HP}}$ Easy | | TimeQA$_{\text{HP}}$ Hard | | SituatedQA |
|---|---|---|---|---|---|---|---|---|---|
| | EM | F1 | EM | F1 | EM | F1 | EM | F1 | EM |
| BigBird (Chen et al., 2021) | 51.2 | 60.3 | 42.4 | 50.9 | 47.6 | 53.7 | 38.8 | 45.9 | - |
| FiD (Chen et al., 2021) | 60.5 | 67.9 | 46.8 | 54.6 | - | - | - | - | - |
| DocTime (Mathur et al., 2022) | **62.4** | **69.6** | 48.2 | 56.4 | - | - | - | - | - |
| BigBird + TML (Chen et al., 2023b) | 55.4 | 63.8 | 45.2 | 55.3 | - | - | - | - | - |
| Longformer + TML (Chen et al., 2023b) | 52.4 | 62.6 | 48.7 | 58.5 | - | - | - | - | - |
| DPR + Query Modified (Zhang and Choi, 2021) | - | - | - | - | - | - | - | - | 23.0 |
| TSM (Cole et al., 2023) | - | - | - | - | - | - | - | - | 24.6 |
| Replicated FiD | 55.3 | 65.1 | 45.2 | 54.5 | 52.7 | 59.7 | 39.7 | 47.0 | 23.7 |
| LongT5 | 55.4 | 64.3 | 49.9 | 58.4 | 53.7 | 61.6 | 45.3 | 52.3 | 24.7 |
| LongT5$_{\text{GNN}}$ | 54.2 | 63.4 | 49.0 | 57.6 | 46.5 | 55.3 | 34.3 | 40.2 | 27.3 |
| LongT5$_{\text{ERR}}$ | 56.9 | 66.2 | **54.0** | **62.0** | **54.9** | **62.9** | **49.3** | **56.1** | **29.0** |
| Replicated FiD (re-annotated) | 64.0 | 68.3 | | | | | | | |
| LongT5$_{\text{ERR}}$ (re-annotated) | **68.8** | **70.4** | | | | | | | |

Table 2: Performance on the test sets. TimeQA$_{\text{HP}}$ denotes the human-paraphrased splits of TimeQA. The highest-performing model is bolded. Confidence intervals for our results are provided in table 7 in appendix A.3.

on TimeQA and SituatedQA, respectively. Appendix A.5 provides other implementation details such as hyperparameters, graph statistics, software versions, and external tool performance.

We perform model selection before evaluating on the test sets, exploring different graph subsets with both the ERR and GNN based fusion approaches introduced in Section 3.2. Table 6 in appendix A.2 shows that the best ERR method uses a document-time-to-question-time (DT2QT) subgraph and the best GNN method uses the full temporal graph by fusing it into the token embedding layer representations of the Transformer model. We hereafter refer to the LongT5 model fused with a DT2QT graph using the ERR method as LongT5$_{\text{ERR}}$, and the LongT5 model fused with a full temporal graph using the GNN method as LongT5$_{\text{GNN}}$.

### 4.4 Main Results

We summarize the performance of baseline models and those trained with our graph fusion methods in Table 2.

**Which baseline models perform best?** On TimeQA, our LongT5 model without temporal graph fusion performs better than or equivalent to all other baseline models across every split and metric except for the Easy split. The best-performing model reported on TimeQA Easy is DocTime. On SituatedQA, LongT5 with no fusion performs as well as the best-reported results on this dataset.

**Which graph fusion methods perform best?** Using LongT5, we consider both of our ERR and GNN fusion methods described in Section 3.2. On

TimeQA, the LongT5$_{\text{GNN}}$ model fails to outperform LongT5 without fusion, while the LongT5$_{\text{ERR}}$ model improves over LongT5 on every split and dataset, exhibiting particularly large gains on the Hard splits. On Situated QA, both LongT5$_{\text{ERR}}$ and LongT5$_{\text{GNN}}$ models improve over the no-fusion LongT5 baseline, with ERR again providing the best performance. The somewhat inconsistent performance of the GNN fusion method across datasets (beneficial on SituatedQA while detrimental on TimeQA) suggests the need for a different GNN design for TimeQA, which we leave to future work.

To explore the differences between LongT5$_{\text{ERR}}$ and LongT5$_{\text{GNN}}$ models, we analyze 20 randomly sampled examples from TimeQA Hard where LongT5$_{\text{ERR}}$ is correct but LongT5$_{\text{GNN}}$ is incorrect. From our manual analysis of these 20 examples, all 20 examples share the same pattern: LongT5$_{\text{GNN}}$ fails to capture explicit temporal expressions in the context and relate them to the question's timeline, which is crucial for deducing the right answer. This suggests that directly embedding pre-computed temporal relations between time nodes into the input is more efficient than implicitly doing so through the GNN, allowing the model to utilize them more easily. Table 14 of appendix A.11 shows three of the analyzed examples.

**Does our approach outperform prior work?** On TimeQA, the LongT5$_{\text{ERR}}$ model achieves a new state-of-the-art on three of the four splits, with TimeQA Easy being the exception. On SituatedQA, the LongT5$_{\text{ERR}}$ model achieves a new state-of-the-art, outperforming the best reported results on this dataset. Our model excels on datasets that require

| Type | Freq | # | Question | Relevant Context | Prediction | Answer |
|------|------|---|----------|------------------|------------|--------|
| False Negative | 40 | 1 | Which team did the player Rivaldo belong to from 1996 to 1997? | After the Olympics …Rivaldo moved to Spain as he joined **Deportivo** La Coruña in La Liga. | Deportivo La Coruña. | Deportivo. |
| | | 2 | What was the position of Albert Reynolds from Dec 1992 to Nov 1994? | Albert Reynolds was an Irish Fianna Fáil politician …, **Leader of Fianna Fáil** from 1992 to 1994…He served as a Teachta Dála (TD) from 1977 to 2002. | Leader of Fianna Fáil. | Teachta Dála. |
| Semantic Understanding | 3 | 3 | Lucas Papademos was an employee for whom from 2010 to 2011? | In 2010 he served as an economic advisor to Greek Prime Minister George Papandreou. | N/A. | Greek Prime Minister George Papandreou. |
| Semantic Understanding & Insufficient Temporal Information | 6 | 4 | Which team did the player Miguel Veloso belong to from 2001 to 2003? | …Veloso started his football career at S.L. **Benfica**, but was rejected for being slightly overweight at the time… | Benfica. | N/A. |
| Semantic Understanding & Temporal Reasoning | 1 | 5 | What operated LMR 57 Lion from 1838 to 1845? | Lion was ordered by the Liverpool & Manchester Railway in October 1837…In 1845 the LMR was absorbed by the Grand Junction Railway (GJR), which in turn was one of the constituents of the **London and North Western Railway** (LNWR) a year later. | London and North Western Railway. | Liverpool & Manchester Railway. |

Table 3: Error categories and examples on the TimeQA Easy development set. Model predictions are in bold. Underlines denote the correct answers.

more temporal reasoning skills, like TimeQA Hard (where our model achieves a 5.8-point higher exact match score than DocTime) and SituatedQA (where our model achieves a 4.3-point higher exact match score than TSM).

Our approach offers further advantages over alternative models due to its simplicity, as summarized in Table 1. The best prior work on TimeQA Easy, DocTime, requires training a temporal dependency parser on additional data, using CAEVO, and modifying the Transformer model's attention mechanism. The best prior work on SituatedQA, TSM, requires an 11-billion parameter T5 model which is pre-trained on the entirety of Wikipedia. In contrast, our approach only uses SUTime to construct a graph, requires only minor adjustments to the Transformer model's input, and outperforms prior work using a model with only 250 million parameters.

**Why does our model not achieve state-of-art performance on TimeQA Easy as it does on other splits and datasets?** On TimeQA Easy, there is a performance gap between our LongT5$_{ERR}$ model and DocTime. Because the DocTime model has not been released we cannot directly compare with its predicted results. Instead, we randomly select 50 errors from our LongT5$_{ERR}$'s output on the TimeQA Easy development set for error analysis. Table 3 shows that most of the errors are false negatives, where the model's predicted answers are typically co-references to the correct answers (as in Table 3 example 1) or additional correct answers that are

applicable in the given context but are not included in the gold annotations (as in Table 3 example 2). The remaining errors are primarily related to semantic understanding, including the inability to accurately identify answer entities (e.g. identifying Greek Prime Minister George Papandreou as an employer in Table 3 example 3), the inability to interpret negation (e.g. in Table 3 example 4, where "rejected" implies that Veloso did not join Benfica), and the inability to reason about time with numerical calculations (e.g. "a year later" in Table 3 example 5 implies 1846). Addressing the semantic understanding errors may require incorporating additional entities and their types into the graphs, as well as better processing of negation information and relative times.

To better understand the extent of false negatives in TimeQA Easy, we re-annotated the 392 test examples where the predictions of the replicated FiD model and our LongT5$_{ERR}$ model are partially correct (i.e., EM = 0 and F1 > 0). We then incorporated additional coreferent mentions into the gold label set for these examples. For instance, if the original gold answer was "University of California," we added its coreferent mention "University of California, Los Angeles" to the gold answers. We then evaluate both the replicated FiD (the best-performing model we can reproduce) and our LongT5$_{ERR}$ model on the re-annotated TimeQA Easy split. The last two rows of Table 2 show that while the exact match score for FiD increases by 8.7, the exact match score for our LongT5$_{ERR}$

| Model | TimeQA Easy | | TimeQA Hard | |
|---|---|---|---|---|
| | EM | F1 | EM | F1 |
| ChatGPT | 41.6 | 56.6 | 25.0 | 32.0 |
| w/ ERR + DT2QT Subgraph | 43.2 | 57.2 | 30.4 | 38.5 |

Table 4: Performance of ChatGPT on 500 sampled test examples, with and without our ERR method.

| LongT5$_{ERR}$ | FiD | Time QA Easy | TimeQA Hard |
|---|---|---|---|
| Correct | Correct | 1427 | 1113 |
| Correct | Wrong | 269 | 494 |
| Wrong | Correct | 243 | 260 |
| Wrong | Wrong | 1082 | 1220 |

Table 5: Comparison of the predictions of LongT5$_{ERR}$ vs. the predictions of FiD.

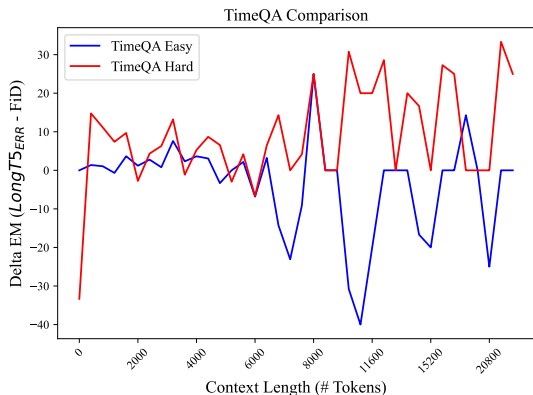

Figure 4: Comparison of the performance on different lengths of context documents.

model increases by 11.9. This suggests that our model may be incurring greater penalties for finding valid coreferent answers than baseline methods.

**Does our ERR method benefit large language models using in-context learning?** We have focused so far on temporal graph fusion when fine-tuning models, but large language models such as ChatGPT (OpenAI, 2022) and LLaMA (Touvron et al., 2023) can achieve impressive performance without additional fine-tuning via in-context learning. Therefore, we tested the performance of ChatGPT (gpt-3.5-turbo) both with and without ERR for fusing the question-time-to-document-time graph. Following previous work (Khattab et al., 2022; Trivedi et al., 2022; Yoran et al., 2023) and considering the cost of ChatGPT's commercial API, we randomly sample 500 examples from TimeQA Easy and TimeQA Hard for evaluation. The prompt format for ChatGPT remains the same as the input format described in Section 3.2.1, except that we concatenate in-context learning few-shot exemplars and task instructions before the input. We evaluate ChatGPT with and without graph fusion using an 8-shot setting. Examples of prompts are provided in Table 11 of Appendix A.8. Table 4 shows that our ERR graph fusion method improves the performance of ChatGPT on TimeQA Easy and particularly on TimeQA Hard. We note that this improvement is possible because our method can easily integrate with state-of-the-art large language models, as our approach to temporal graph fusion modifies only the input sequence. Prior work which relies on modifying attention mechanisms or adding graph neural network layers is incompatible with this in-context learning setting.

## 5 Analysis

In this section, we analyze our LongT5$_{ERR}$ model on the TimeQA development set.

**How do predictions differ compared to FiD?** We compare the predictions of LongT5$_{ERR}$ to the replicated FiD model in Table 5. While LongT5$_{ERR}$ and FiD both correct about the same number of

each other's errors on TimeQA Easy (269 vs. 243), LongT5$_{ERR}$ corrects many more of FiD's errors than the reverse on TimeQA Hard (494 vs. 260). To further analyze these cases, we sampled 10 errors from the set where LongT5$_{ERR}$ was correct while FiD was incorrect as well as the set where the FiD was correct while LongT5$_{ERR}$ was incorrect. We did this across both TimeQA Easy and TimeQA Hard, totaling 40 examples. Among the 20 examples in which LongT5$_{ERR}$ was correct and FiD was incorrect, 17 have node markers near the answers in the ERR input sequence, and the most frequent ones are <included by> and <overlap>. The remaining 3 examples have unanswerable questions. In the examples in which FiD was correct while LongT5$_{ERR}$ was incorrect, we observe that 13 examples are additional correct answers (i.e., false negatives), while the other 7 examples are semantic understanding errors similar to those discussed previously. These results suggest that our ERR graph fusion approach is providing the model with useful targets for attention which allow it to produce more correct answers.

**How does the length of the document affect performance?** We compare the performance of LongT5$_{ERR}$ to the replicated FiD model on various document lengths, as depicted in Figure 4.

LongT5$_{ERR}$ performs less competitively than FiD on the Easy split for longer documents. This could be attributed to a high frequency of false negatives in LongT5$_{ERR}$, as discussed previously. Additionally, it could be that LongT5$_{ERR}$ is less efficient at string matching on longer documents than FiD. Most of the question times in the Easy split are explicitly mentioned in the context document, which can be solved via string matching rather than temporal reasoning. However, our LongT5$_{ERR}$ model shows a substantial improvement on TimeQA Hard, outperforming the FiD model across most lengths.

## 6  Conclusion

In this paper, we compared different methods for fusing temporal graphs into Transformer models for time-sensitive question answering. We found that our ERR method, which fuses the temporal graph into the Transformer model's input, outperforms GNN-based fusion as well as attention-based fusion models. We also showed that, unlike prior work on temporal graph fusion, our approach is compatible with in-context learning and yields improvements when applied to large language models such as ChatGPT. Our work establishes a promising research direction on fusing structured graphs with the inputs of Transformer models. In future work, we intend to use better-performing information extraction systems to construct our temporal graphs, enhance our approach by adding entities and entity type information to the graphs, and extend our method to spatial reasoning.

## Limitations

We use CAEVO and SUTime to construct temporal graphs because of their scalability and availability. Using more accurate neural network-based temporal information extraction tools may provide better temporal graphs but may require domain adaptation and retraining.

While we did not find graph convolutions to yield successful fusion on TimeQA, we mainly explored variations of such methods proposed by prior work. We also did not explore self-attention based fusion methods, as preliminary experiments with those methods yielded no gains, and DocTime provided an instance of such methods for comparison. But there may exist other variations of graph convolution and self-attention based fusion methods beyond those used in prior work that would make such methods more competitive with our

input-based approach to fusion.

We also did not deeply explore the places where graph neural networks failed. For example, the graph convolution over the final layer contextualized embeddings using the full temporal graph yielded performance much lower than all other graph convolution variants we explored. We limited our exploration of failures to the more successful explicit edge representation models.

## Ethics Statement

Wikipedia contains certain biases (Falenska and Çetinoğlu, 2021), and we use data from Wikipedia to train our model, thus we are potentially introducing similar biases into our models.

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

# A Appendix

## A.1 Transitivity rules

Figure 5 shows the standard temporal transitivity rules we apply to infer relations from paths through the temporal graph.

## A.2 Model Selection

For the ERR method introduced in Section 3.2.1, we consider a document-time-to-question-time (DT2QT) subgraph that contains all the edges connecting document-time nodes to the question-time node, and a document-time-event-to-question-time (DTE2QT) subgraph that builds on the document-time-to-question-time subgraph by adding all the edges from the document-event nodes to the question-time node. We do not fuse the entire temporal graph with ERR because doing so would significantly increase the length of the input. On average, this fusion will add a number of tokens equal to twice the number of edges because each edge needs to be represented by two special markers, e.g., <before> 2009 </before>.

For the GNN method, we consider both the entire temporal graph, and an "all time graph" derived from the DT2QT subgraph, with additional edges for the temporal relations between each time expression node and all other nodes.

We also consider whether to use all six relations or only three relations, merging (BEFORE, AFTER, SIMULTANEOUS, INCLUDES, INCLUDED BY, OVERLAP) into (BEFORE, AFTER, and OVERLAP).

We used the TimeQA Easy and Hard development sets for model selections. Table 6 shows the results of model selection. For ERR, merging the relations down to 3 slightly decreases performance on TimeQA Hard, the DT2QT graph is slightly better for TimeQA Hard, and the DTE2QT graph is slightly better for TimeQA Easy. Since the DT2QT is simpler, we select this model as our ERR model.

For GNN, no models outperform LongT5 alone, regardless of whether the GNN was applied to the token embeddings or the final contextualized embeddings, and whether applied to simpler or more complex temporal graphs.

## A.3 Confidence Intervals

We use the bootstrap resampling method to construct 95% confidence intervals on the test sets. The results are shown in Table 7.

## A.4 Software Licenses

We train question answering systems to answer English Wikipedia time-sensitive questions. Upon publication, we will release our code under the

| | before | after | includes | included by | simultaneous | overlap |
|---|---|---|---|---|---|---|
| **before** | before | $\times$ | before | $\times$ | before | $\times$ |
| **after** | $\times$ | after | after | $\times$ | after | $\times$ |
| **includes** | $\times$ | $\times$ | includes | $\times$ | includes | $\times$ |
| **included by** | before | after | $\times$ | included by | included by | $\times$ |
| **simultaneous** | before | after | includes | included by | simultaneous | overlap |
| **overlap** | $\times$ | $\times$ | $\times$ | overlap | overlap | $\times$ |

Figure 5: Transitivity rules.

| Model | TimeQA Dev | | | |
|---|---|---|---|---|
| | Easy | | Hard | |
| | EM | F1 | EM | F1 |
| LongT5 | 55.2 | 63.5 | 48.1 | 56.4 |
| LongT5 + ERR + DT2QT Subgraph | 56.1 | 65.1 | **52.1** | **60.5** |
| LongT5 + ERR + DT2QT Subgraph (merge to 3 relations) | 56.7 | 65.3 | 51.9 | 60.1 |
| LongT5 + ERR + DTE2QT Subgraph | **57.0** | **65.6** | 50.6 | 59.0 |
| LongT5 + GNN Token Embed + All Time Graph | 55.3 | 64.1 | 47.8 | 56.3 |
| LongT5 + GNN Token Embed + Temporal Graph | 55.5 | 64.3 | 48.1 | 56.8 |
| LongT5 + GNN Context Embed + All Time Graph | 11.5 | 11.5 | 13.4 | 13.4 |
| LongT5 + GNN Context Embed + Temporal Graph | 53.9 | 62.5 | 48.8 | 57.6 |

Table 6: Performance on TimeQA development datasets. The highest-performing model is bolded. DT2QT Subgraph is the document-time-to-question-time subgraph. DTE2QT Subgraph is the document-time-event-to-question-time subgraph.

| Models | TimeQA Test | | | | SituatedQA Test |
|---|---|---|---|---|---|
| | Easy | Hard | HP Easy | HP Hard | |
| | EM / F1 | EM / F1 | EM / F1 | EM / F1 | F1 |
| Replicated FiD | 55.3 / 65.1 | 45.2 / 54.5 | 52.7 / 59.7 | 39.7 / 47.0 | - |
| | [53.9, 56.6] / [63.9, 66.3] | [43.8, 46.6] / [53.2, 55.7] | [50.2, 55.2] / [57.4, 61.9] | [37.3, 42.2] / [44.7, 49.4] | |
| LongT5 | 55.4 / 64.3 | 49.9 / 58.4 | 53.7 / 61.6 | 45.3 / 52.3 | 24.7 |
| | [54.1, 56.7] / [63.1, 65.5] | [48.5, 51.2] / [57.2, 59.7] | [51.3, 55.9] / [59.3, 63.7] | [42.8, 47.6] / [50.0, 54.6] | [23.2, 26.4] |
| LongT5$_{GNN}$ | 54.2 / 63.4 | 49.0 / 57.6 | 46.5 / 55.3 | 34.3 / 40.2 | 27.3 |
| | [52.8, 55.6] / [62.1, 64.5] | [47.6, 50.3] / [56.4, 58.8] | [44.1, 48.8] / [53.1, 57.4] | [32.0, 36.3] / [38.0, 42.3] | [25.8, 29.3] |
| LongT5$_{ERR}$ | **56.9 / 66.2** | **54.0 / 62.0** | **54.9 / 62.9** | **49.3 / 56.1** | **29.0** |
| | [55.5, 58.3] / [65.0, 67.5] | [52.6, 55.3] / [60.8, 63.2] | [52.5, 57.1] / [60.8, 65.0] | [46.9, 51.5] / [53.9, 58.1] | [27.3, 30.6] |

Table 7: Confidence intervals on the test sets, generated via bootstrap resampling. HP = Human-Paraphrased.

MIT license. We list below the licenses of the data, software, and models we used. Our use is consistent with their intended uses.

- Stanford CoreNLP: GNU General Public License Version 3 [6].
- CAEVO, Long-t5-tglobal-base model, HuggingFace Transformers: Apache License, Version 2.0 [7].
- TimeQA dataset: BSD 3-Clause License [8].
- Pytorch: BSD-style license [9].

## A.5 Implementation Details

We use SUTime from Stanford CoreNLP V4.5.0 (Manning et al., 2014) to identify and normalize time expressions. We use the LongT5 implementation from Huggingface Transformers V4.20.1 (Wolf et al., 2020) and the LongT5 base model (250 million parameters) with transient-global attention checkpoint (long-t5-tglobal-base) from Google as the starting point for training. We follow the code provided in the FiD Github repository[10] to preprocess the Natural Questions dataset. We create new tokens in the tokenizer and model vocabulary for special marker tokens such as <e>. We tune the learning rate $r \in \{1 \times 10^{-5}, 2 \times 10^{-5}, 3 \times 10^{-5}, 4 \times 10^{-5}, 5 \times 10^{-5}\}$, batch size $b \in \{4, 8, 16, 32\}$, and number of RelGraphConv layers $l \in \{1, 3, 6\}$ on the development set of TimeQA and use early stopping to monitor the exact match metrics. We use $ReLU$ as activation functions in RelGraphConv layers. We set the hidden state size of RelGraphConv layers to the same as LongT5-Base. All experiments are conducted on 4 Nvidia A6000 GPUs.

The total GPU hours of experiments are around 160. We will make our code publicly available upon publication of the paper[11].

**Graph Statistics** The constructed graphs statistics are presented in Table 8.

**Performance of the External Tools** We use two tools: SUTime from Stanford CoreNLP and CAEVO. CAEVO's reported precision is 0.92 for event-time relations and 0.88 for time-time relations. The accuracy of SUTime in recognizing time expression types and values is 0.96 and 0.82, respectively.

## A.6 Datasets

Table 9 presents the statistics of the datasets used for the experiments. Table 10 shows the input length of the LongT5 model before and after fusing the DT2QT sub-graph using the ERR method.

## A.7 Temporal Relations

We visualize the temporal relations between the time intervals in Figure 6.

## A.8 ChatGPT Prompts

We present the ChatGPT prompts we used in Table Table 11. Considering the overall input length constraint, for context documents in in-context learning examples, we use sentences containing answers as context for questions that have answers. For questions without answers, we randomly sample a sentence from the original context document as the context.

## A.9 GPT-4 Results

Following the in-context learning setup described in Section 4.4, we replicate the experiment using

---

[6] www.gnu.org/licenses/gpl-3.0.html
[7] www.apache.org/licenses/LICENSE-2.0
[8] /opensource.org/licenses/BSD-3-Clause
[9] github.com/pytorch/pytorch/blob/master/LICENSE
[10] https://github.com/facebookresearch/FiD

[11] https://github.com/IntelLabs/multimodal_cognitive_ai/tree/main/Fusing_Temporal_Graphs

| Dataset | Split | Avg. # Nodes | Avg. # Edges | In/Out Degree |
|---|---|---|---|---|
| TimeQA Easy | Train | 161 | 323 | 2/2 |
| TimeQA Easy | Dev | 163 | 326 | 2/2 |
| TimeQA Easy | Test | 163 | 324 | 2/2 |
| TimeQA Hard | Train | 160 | 321 | 2/2 |
| TimeQA Hard | Dev | 162 | 325 | 2/2 |
| TimeQA Hard | Test | 162 | 323 | 2/2 |
| TimeQA HP Easy | Train | 163 | 317 | 2/2 |
| TimeQA HP Easy | Test | 194 | 376 | 2/2 |
| TimeQA HP Hard | Train | 163 | 317 | 2/2 |
| TimeQA HP Hard | Test | 194 | 376 | 2/2 |
| SituatedQA TEMP | Train | 214 | 213 | 1/1 |
| SituatedQA TEMP | Dev | 210 | 209 | 1/1 |
| SituatedQA TEMP | Test | 213 | 212 | 1/1 |

Table 8: The statistics of constructed graphs. HP = Human-Paraphrased.

| Dataset | Split | # Questions | # Answerable | # Unanswerable | Avg. # Tokens |
|---|---|---|---|---|---|
| TimeQA Easy | Train | 14308 | 12532 | 1776 | 2566 |
| TimeQA Easy | Development | 3021 | 2674 | 347 | 2635 |
| TimeQA Easy | Test | 2997 | 2613 | 384 | 2634 |
| TimeQA Hard | Train | 14681 | 12532 | 2149 | 2560 |
| TimeQA Hard | Development | 3087 | 2674 | 413 | 2636 |
| TimeQA Hard | Test | 3078 | 2613 | 465 | 2636 |
| Human-Paraphrased TimeQA Easy | Train | 1171 | 982 | 189 | 2907 |
| Human-Paraphrased TimeQA Easy | Test | 1171 | 982 | 189 | 2907 |
| Human-Paraphrased TimeQA Hard | Train | 1171 | 982 | 189 | 2907 |
| Human-Paraphrased TimeQA Hard | Test | 1171 | 982 | 189 | 2907 |
| SituatedQA TEMP | Train | 6009 | 6009 | 0 | 9923[*] |
| SituatedQA TEMP | Dev | 3423 | 3423 | 0 | 9924[*] |
| SituatedQA TEMP | Test | 2795 | 2795 | 0 | 9922[*] |

Table 9: Statistics for the four datasets. Avg. # Tokens is the average number of tokens in the context document. We use the LongT5 model's tokenizer to tokenize context documents. [*]The length is based on the retrieved top 100 paragraphs from Wikipedia, which serve as a context document for each question. If the total length of a context document exceeds 10,000 tokens, we truncate the context accordingly.

the GPT-4 model. However, due to the costs of calling the commercial GPT-4 API, we only randomly sampled 100 examples each from TimeQA Easy and TimeQA Hard, resulting in a total of 200 samples for the experiment. The results are shown in Table 12. ERR method achieves similar performance improvements with GPT-4 as it does with ChatGPT (gpt-3.5-turbo).

## A.10 Additional Error Analysis

Table 13 presents error examples, in addition to the examples shown in Table 3.

## A.11 Analysis of the GNN Fusion Method

Table 14 shows examples from TimeQA Hard where LongT5$_{ERR}$ is correct but LongT5$_{GNN}$ is incorrect.

## A.12 Other Experimented Methods

The following are some of the methods we have tried but yielded lower performance than our main methods reported in Table 6.

1. We tried to use the coreference resolution tools to process the context documents and then construct and fuse temporal graphs based on the processed text, but we found that the coreference resolution preprocessing hurt the performance of the models.

2. We tried to fuse constructed temporal graphs into FiD models using relation-aware attention (Wang et al., 2020), but we found that the fused FiD models performed almost the same as the non-fused FiD models.

3. Our preliminary study found that the performance of the models can be significantly improved if only the most relevant paragraph in the context document is used as the context.

| Dataset | # Tokens Pre-ERR | # Tokens Post-ERR |
|---------|------------------|-------------------|
| TimeQA | 2617 | 2862 |
| SituatedQA | 9923 | 11515 |

Table 10: Average LongT5 input length before and after using the ERR method to fuse DT2QT subgraphs.

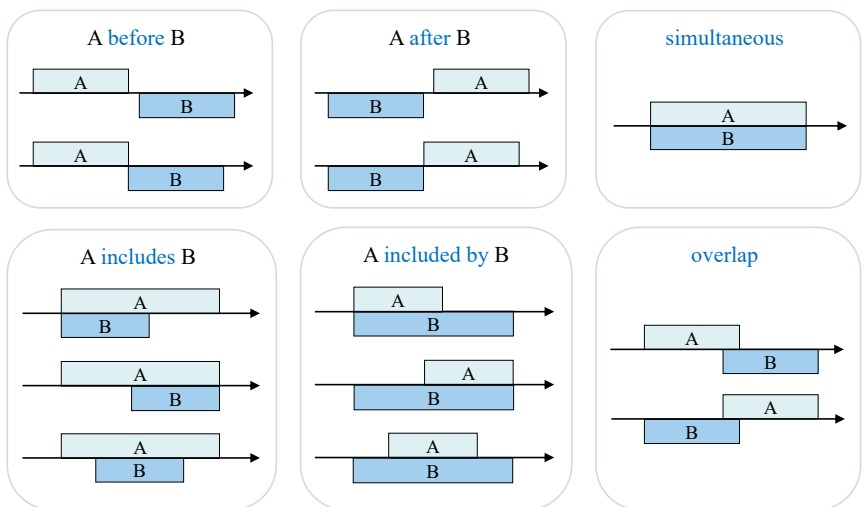

Figure 6: Visualization of temporal relations.

---

**W/O Graph Fusion**

---

**Instruction**: Answer the question based on the given context. If there is no answer, answer "no answer."

**Context**: In the 1955 general election, Cunningham was chosen as the new Ulster Unionist MP for South Antrim. He was a delegate to the Council of Europe and Western European Union Parliamentary Assembly from 1956 to 1959.
**Question**: Which position did Knox Cunningham hold before Apr 1956?
**Answer**: Ulster Unionist MP for South Antrim

**Context**: Broughton would go on to teach at Amherst College, Bryn Mawr College (1928-1965) and, later, serve as George L.
**Question**: Thomas Robert Shannon Broughton was an employee for whom before Jun 1926?
**Answer**: no answer

---

**W/ Document-Time-to-Question-Time Subgraphs Fused Using ERR Method**

---

**Instruction**: Answer the question based on the given context. If there is no answer, answer "no answer." The temporal relations between the times mentioned in the context and the question are represented using XML-style tags.

**Context**: In the <included by>1955</included by> general election, Cunningham was chosen as the new Ulster Unionist MP for South Antrim. He was a delegate to the Council of Europe and Western European Union Parliamentary Assembly <overlap>from 1956 to 1959</overlap>.
**Question**: Which position did Knox Cunningham hold <question time>before Apr 1956</question time>?
**Answer**: Ulster Unionist MP for South Antrim

**Context**: Broughton would go on to teach at Amherst College, Bryn Mawr College (<after>1928-1965</after>) and, later, serve as George L.
**Question**: Thomas Robert Shannon Broughton was an employee for whom <question time>before Jun 1926</question time>?
**Answer**: no answer

---

Table 11: Examples of ChatGPT prompts.

| Model | TimeQA Easy | | TimeQA Hard | |
|-------|-----|-----|-----|-----|
| | EM | F1 | EM | F1 |
| GPT-4 | 52.0 | 62.1 | 48.0 | 56.7 |
| w/ ERR + DT2QT Subgraph | 53.0 | 64.0 | 55.0 | 63.5 |

Table 12: Performance of GPT-4 on 100 randomly sampled test examples, with and without our ERR method.

| Type | Freq | # | Question | Relevant Context | Prediction | Answer |
|------|------|---|----------|------------------|------------|--------|
| False Negative | 40 | 1 | Where was Richard G. Hovannisian educated from 1954 to 1958? | Hovannisian received his B.A. in history (1954) from the **University of California, Berkeley**, and his M.A. in history (1958) and his Ph.D. (1966) from University of California, Los Angeles (UCLA). | University of California, Berkeley | University of California, Los Angeles (UCLA) |
| | | 2 | Which team did the player Simone Verdi belong to from 2014 to 2015? | On 18 June 2014, the co-ownership between Torino and Milan was renewed for a third year, with Verdi loaned to **Empoli** again. | Empoli | Milan |
| | | 3 | Segenet Kelemu was an employee for whom from Nov 2013 to Nov 2014? | In 2013, Kelemu joined the **Alliance for a Green Revolution in Africa** (AGRA) as Vice President for Programs for about a year. In November 2013, Kelemu became the Director General of the International Centre of Insect Physiology and Ecology (icipe) . . . | Alliance for a Green Revolution in Africa | International Centre of Insect Physiology and Ecology |
| Semantic Understanding | 3 | 4 | Donald Thobega played for which team in 2006? | . . . Between 1999 and 2006 he won a total of five caps for the Botswana national football team. | N/A | Botswana national football team |
| | | 5 | Which school did Alexander Dvorkin go to from 1975 to 1978? | . . . in 1972, he became a student in the Faculty of Russian Language and Literature of **Moscow Pedagogical Institute** . . . consequently, in the autumn of 1975, Dvorkin was expelled from the third year of the institute . . . | Moscow Pedagogical Institute | N/A |
| Semantic Understanding & Insufficient Temporal Information | 6 | 6 | What was the capital of Kolomna from 1929 to Apr 2017? | Within the framework of administrative divisions, Kolomna serves as the administrative center of Kolomensky District . . . | N/A | Kolomensky District |
| | | 7 | Where did Ryan Gander work from 2007 to 2015? | . . . In 2004, he was made Cocheme Fellow at Byam Shaw School of Art, London . . . | N/A | London |

Table 13: Error categories and examples on the TimeQA Easy development set. Model predictions are in bold. Underlines denote the correct answers.

Thus, we tried to train a cross-encoder-based ranker to rank the paragraphs in the context documents, but we found that the pipeline approach using a cross-encoder as a paragraphs ranker and a LongT5 as a reader was not as good as using LongT5 directly to generate answers end-to-end.

**Question**: What position did Roland Burris take in May 1989?
**LongT5$_{ERR}$ Prediction**: Illinois Comptroller
**LongT5$_{GNN}$ Prediction**: Illinois Attorney General
**Context**: ... In 1978, Burris was the first African-American elected to statewide office in Illinois, when he was elected Illinois Comptroller. He served in that office until his election as Illinois Attorney General in 1990 ...
**Annotated explanation**: LongT5$_{GNN}$ prediction of "Illinois Attorney General" is based on the fact that Burris was elected to this position in 1990. However, it failed to correctly link the timeline in the question "May 1989" to the context.

**Question**: Where was Joe Burrow educated in Feb 2015?
**LongT5$_{ERR}$ Prediction**: Ohio State
**LongT5$_{GNN}$ Prediction**: Athens High School
**Context**: ... Burrow attended Athens High School (2011–14) in The Plains, Ohio ... After redshirting his first year at Ohio State in 2015, Burrow spent the next two years as a backup to J. T. Barrett ...
**Annotated explanation**: LongT5$_{GNN}$'s guess of "Athens High School" is based on Burrow's education at Athens High School in The Plains, Ohio. However, it failed to discern how "Feb 2015" in the question aligns with the timeline in the context. As evident, Burrow was at Ohio State in 2015.

**Question**: Which school did Kyriakos Mitsotakis attend in Feb 1985?
**LongT5$_{ERR}$ Prediction**: Athens College
**LongT5$_{GNN}$ Prediction**: Harvard University
**Context**: ... In 1986, he graduated from Athens College. From 1986 to 1990, he attended Harvard University and earned a bachelor's degree in social studies, receiving the Hoopes Prize ...
**Annotated explanation**: LongT5$_{GNN}$'s prediction "Harvard University" is based on Mitsotakis attending Harvard from 1986 to 1990. Yet, the timeline "Feb 1985" in the question doesn't match the context. From the context, it's clear Mitsotakis graduated from Athens College in 1986, implying he was studying there in 1985.

Table 14: Examples where LongT5$_{ERR}$ is correct, but LongT5$_{GNN}$ is incorrect.