# OpenReview forum: "Fusing Temporal Graphs into Transformers for Time-Sensitive Question Answering"
_EMNLP/2023/Conference — EMNLP 2023 Findings_

### Official Review · Reviewer_ay7N · 2023-08-05

**Soundness:** 3

**Excitement:**

4: Strong: This paper deepens the understanding of some phenomenon or lowers the barriers to an existing research direction.

**Paper Topic And Main Contributions:**

The paper explores temporal reasoning in answering time-sensitive questions from long documents. The authors investigate whether large language models can perform this reasoning solely using text or if they can benefit from additional temporal information. They apply existing temporal information extraction systems to construct temporal graphs and propose different approaches for fusing them into Transformer models. Experimental results show that their approach enhances the temporal reasoning capabilities of Transformer models and outperforms other graph convolution-based approaches, achieving state-of-the-art performance on SituatedQA and TimeQA datasets.

The main contributions of the paper are as follows:

1. Applying existing temporal information extraction systems to construct temporal graphs of events, times, and temporal relations in questions and documents.

2. Investigating different approaches for fusing these temporal graphs into Transformer models.

3. Proposing an approach called Explicit Edge Representation Fusion (ERR) to directly fuse the temporal graphs into the input text of the Transformer model.

4. Another approach is Graph neural network fusion: This approach utilizes a Graph Neural Network (GNN) to fuse the temporal graphs into the token embeddings or the last hidden layer representations of the Transformer model.

5. Shows that the proposed ERR approach enhances the temporal reasoning capabilities of Transformer models and outperforms various graph convolution-based approaches, achieving state-of-the-art performance on SituatedQA and three splits of TimeQA datasets.


The experimental results and the proposed approach to achieving state-of-the-art performance are summarized as follows:

1. Baseline Models: The LongT5 model without temporal graph fusion performs better than or equivalent to all other baseline models on TimeQA and SituatedQA datasets, except for the Easy split. The best-performing model on TimeQA Easy is DocTime.

2. Graph Fusion Methods:  On TimeQA, the LongT5GNN model does not outperform LongT5 without fusion. However, the LongT5ERR model improves over LongT5 on every split and dataset, particularly on the Hard splits. On SituatedQA, both LongT5ERR and LongT5GNN models improve over the no-fusion LongT5 baseline, with ERR providing the best performance.

**Questions For The Authors:**

Q1: Are the reported results obtained from a singular program run, or have multiple runs been conducted? In a different run, you may get a lower or higher result compared to other baselines.

Q2: If we have the labels such as <before> or <after>, why must we pass this to a language model for the final reasoning? Cannot we do logical computations over these to find the answer? For example, if we know that event A is before event B and event C is after event B then we can get that event C is after event A too.

Q3: This concept merits exploration beyond its current application in temporal reasoning, showcasing potential generality and utility in broader reasoning contexts. Given its effectiveness in temporal reasoning, one may speculate about the feasibility of applying the same approach in other logical reasoning domains, such as spatial reasoning.

**Reasons To Accept:**

S1. The proposed approach achieves state-of-the-art performance on TimeQA and SituatedQA datasets.

S2: The proposed approach offers advantages over alternative models in simplicity. It requires only minor adjustments to the Transformer model's input. It outperforms prior work using a model with only 250 million parameters, while other models require additional data, complex parsers, or large pre-trained models.

S3:  The paper also shows the advantages of the ERR approach in the ChatGPT model.

S4: The writing was clear and easy to follow.

**Reasons To Reject:**

W1: The paper presents two distinct methodologies, ERR and GNN, to integrate graph knowledge into language models. However, an explicit comparative analysis between these approaches needs to be included. The rationale behind the superior performance of ERR vs. GNN remains unexplored, even though ERR employs a more straightforward data infusion technique than GNN. Investigating the reasons behind this performance disparity could shed valuable light on these approaches' strengths and weaknesses, contributing to a more comprehensive understanding of their effectiveness.

W2: Despite the paper's strength lying in its simplicity and ease of application to other benchmarks, a more comprehensive analysis could be achieved by comparing this approach with others that incorporate textual data into large language models. For instance, employing code execution might offer deeper insights and further enrich the investigation.

W3: Acknowledging that ChatGPT exhibits certain limitations in reasoning, some of which have been addressed in models like InbdtructGPT, Flan-T5, PaLM2, or GPT4(ref: Towards Benchmarking and Improving the Temporal Reasoning Capability
of Large Language Models, 2023), choosing ChatGPT as the representative language model for this study may not be the optimal choice. It is plausible to hypothesize that more advanced language models, such as GPT4, might already possess the capacity for temporal reasoning without the explicit infusion of graph data. Nevertheless, the proposed approach could still offer valuable enhancements, even for powerful language models like GPT4.

**Reproducibility:**

4: Could mostly reproduce the results, but there may be some variation because of sample variance or minor variations in their interpretation of the protocol or method.

**Reviewer Confidence:**

4: Quite sure. I tried to check the important points carefully. It's unlikely, though conceivable, that I missed something that should affect my ratings.

---

> ### Author Rebuttal · Authors · 2023-08-29
>
> Dear reviewer ay7N,
>
> Thank you for your comprehensive review, which we found insightful in both its depth and constructive details. We are grateful for your recognition of our work's contributions, especially the ERR method and its state-of-the-art performance. Your remarks on its simplicity, integration with ChatGPT, and the clarity of our writing are particularly appreciated. We address your constructive comments and feedback raised in the review below.
>
> > W1: The paper presents two distinct methodologies, ERR and GNN, to integrate graph knowledge into language models. However, an explicit comparative analysis between these approaches needs to be included. The rationale behind the superior performance of ERR vs. GNN remains unexplored, even though ERR employs a more straightforward data infusion technique than GNN. Investigating the reasons behind this performance disparity could shed valuable light on these approaches' strengths and weaknesses, contributing to a more comprehensive understanding of their effectiveness.
>
> We appreciate this suggestion and agree that a qualitative comparison of the GNN approach vs. the ERR approach would be a valuable contribution to our paper. We will add the following analysis to the final version of our paper to address this question.
>
> To better understand the differences in performance between the GNN and ERR methods, we analyzed 20 randomly sampled samples from TimeQA Hard where the ERR method was correct but the GNN method was incorrect. From our manual analysis of these 20 samples, we found that all 20 samples shared the same pattern: the GNN method failed to capture explicit temporal expressions in the context and relate them to the question's timeline, which is crucial for deducing the right answer. This suggests that directly embedding pre-computed relationships between time nodes into the input is more efficient than implicitly doing so through the GNN, allowing the model to utilize them more easily.
>
> To illustrate, we provide three of our analyzed samples below with an annotated explanation for the error:
>
> Example 1:
>
> - Question: What position did Roland Burris take in May 1989?
> - ERR prediction (correct answer): Illinois Comptroller
> - GNN prediction (wrong answer): Illinois Attorney General
> - Context: ... In 1978, Burris was the first African-American elected to statewide office in Illinois, when he was elected **Illinois Comptroller**. He served in that office until his election as **Illinois Attorney General** in 1990 ...
> - Annotated explanation: The GNN prediction of "Illinois Attorney General" is based on the fact that Burris was elected to this position in 1990. However, it failed to correctly link the timeline in the question "May 1989" to the context.
>
> Example 2:
>
> - Question: Where was Joe Burrow educated in Feb 2015?
> - ERR prediction (correct answer): Ohio State
> - GNN prediction (wrong answer): Athens High School
> - Context: ... Burrow attended **Athens High School** (2011–14) in The Plains, Ohio ... After redshirting his first year at **Ohio State** in 2015, Burrow spent the next two years as a backup to J. T. Barrett ...
> - Annotated explanation: The GNN's guess of "Athens High School" is based on Burrow's education at Athens High School in The Plains, Ohio. But it failed to discern how "Feb 2015" in the question aligns with the timeline in the context. As evident, Burrow was at Ohio State in 2015.
>
> Example 3:
>
> - Question: Which school did Kyriakos Mitsotakis attend in Feb 1985?
> - ERR prediction (correct answer): Athens College
> - GNN prediction (wrong answer): Harvard University
> - Context: ... In 1986, he graduated from **Athens College**. From 1986 to 1990, he attended **Harvard University** and earned a bachelor's degree in social studies, receiving the Hoopes Prize ...
> - Annotated explanation: GNN's prediction "Harvard University" is based on Mitsotakis attending Harvard from 1986 to 1990. Yet, the timeline "Feb 1985" in the question doesn't match the context. From the context, it's clear Mitsotakis graduated from Athens College in 1986, implying he was studying there in 1985.
>
> > W2: Despite the paper's strength lying in its simplicity and ease of application to other benchmarks, a more comprehensive analysis could be achieved by comparing this approach with others that incorporate textual data into large language models. For instance, employing code execution, as explored in the paper "Unlocking Temporal Question Answering for Large Language Models Using Code Execution, 2023," might offer deeper insights and further enrich the investigation.
>
> Thank you for acknowledging the simplicity and ease of application of our work, and for providing this relevant reference. Unfortunately, we found that the code and data associated with this paper are unavailable, which makes a direct comparison to it in our experimental setting impossible. We also noticed that it was first uploaded to arXiv on 24 May 2023, which classifies it as contemporaneous to our submission per ACL policy. Nevertheless, we found this paper very interesting and will include the following discussion in Section 2 of our final manuscript.
>
> “Li et al., (2023) primarily focus on exploring large language models for information extraction in structured temporal contexts. They represent the extracted time-sensitive information in code, and then execute Python scripts to derive the answers. In contrast, we adopt the original TimeQA setup, where the context document is long and unstructured. This poses more challenges in information extraction and involves more complex reasoning, which motivates our integration of existing temporal information extraction systems with transformer-based language models.”
>
> Xingxuan Li, Liying Cheng, Qingyu Tan, Hwee Tou Ng, Shafiq Joty, Lidong Bing, Unlocking Temporal Question Answering for Large Language Models Using Code Execution, 2023
>
> > W3: Acknowledging that ChatGPT exhibits certain limitations in reasoning, some of which have been addressed in models like InbdtructGPT, Flan-T5, PaLM2, or GPT4(ref: Towards Benchmarking and Improving the Temporal Reasoning Capability of Large Language Models, 2023), choosing ChatGPT as the representative language model for this study may not be the optimal choice. It is plausible to hypothesize that more advanced language models, such as GPT4, might already possess the capacity for temporal reasoning without the explicit infusion of graph data. Nevertheless, the proposed approach could still offer valuable enhancements, even for powerful language models like GPT4.
>
> We appreciate this suggestion and the relevant reference. During the rebuttal period, we repeated our previous ChatGPT experiments using GPT-4. The results are presented in the tables below. We observed that our ERR method achieved similar performance improvements with GPT-4 as it did with ChatGPT. Specifically, it improved the model's performance on both TimeQA Easy and particularly on TimeQA Hard.
>
> |                            | Metric | TimeQA Easy | TimeQA Hard |
> |:--------------------------:|:------:|:-----------:|:-----------:|
> | **GPT-4**                |   EM   |     52.0    |    48.0    |
> |                                  |   F1   |     62.1   |   56.7     |
> | **w/ ERR + DT2QT Subgraph**|   EM   |   **53.0**     |    **55.0**   |
> |                                                    |   F1   |     **64.0**   |   **63.5**    |
>
>
> For this experiment, we used an identical setting as the ChatGPT experiment described in our paper. However, we randomly sampled 100 examples each from TimeQA Easy and TimeQA Hard (resulting in a total of 200 samples for the experiment) due to the cost of the commercial GPT-4 API. We will update the final version of our paper to include these new results.
>
> > Q1: Are the reported results obtained from a singular program run, or have multiple runs been conducted? In a different run, you may get a lower or higher result compared to other baselines.
>
> Thank you for this question. In Table 2, we report the performance of our single best-performing model after hyperparameter tuning experiments. However, we agree that it is important to evaluate the variability in the performance of individual models. To accurately assess this aspect of performance, we followed the method proposed by Dror et al. (2018), employing bootstrap resampling on the test set to obtain confidence intervals of our model's performance. This technique is statistically more appropriate for comparing performance on a test set than running multiple times with different random seeds. We provide the 95% confidence intervals on our models’ performance in Table 7 of Appendix A.3.
>
> Nonetheless, we are grateful for the reviewers' suggestions. We will include the average performance over multiple runs as an additional reference in the final version's appendix.
>
> [Dror et al., 2018] Rotem Dror, Gili Baumer, Segev Shlomov, and Roi Reichart. 2018. The hitchhiker’s guide to testing statistical significance in natural language processing.
>
> > Q2: If we have the labels such as <before> or <after>, why must we pass this to a language model for the final reasoning? Cannot we do logical computations over these to find the answer? For example, if we know that event A is before event B and event C is after event B then we can get that event C is after event A too.
>
> Thank you for this insightful question. The reasoning required by long-document time-sensitive question answering is often complex, which makes it challenging to obtain an answer solely through straightforward temporal logical reasoning. A language model is still essential to understand the text and integrate the information about the question distributed throughout the long document. Our work aims to assist the model in one specific facet of this reasoning: its temporal reasoning.
>
> To illustrate, consider the example shown in Figure 1 of our paper: "What was George Washington’s position between 1776 - 1780?". The reasoning needed to derive the correct answer includes:
>
> 1. Identifying and understanding the distributed information about George Washington’s career progression from the text (e.g., “Congress created the Continental Army on June 14, 1775, and Samuel and John Adams nominated Washington to become its Commander-in-Chief.")
> 2. Recognizing events or facts mentioned in the text related to George Washington during the period 1776-1780. For instance, the document mentions Washington bidding farewell to his officers at Fraunces Tavern in December 1783 and resigning his commission days later.
> 3. Inferring Washington's activities and roles during this period, and concluding that he primarily served as the Commander-in-Chief of the Continental Army.
> 4. Establishing a timeline of these events to ensure that the answer aligns with the queried timeframe. While Washington was elected as the Chancellor of the College of William & Mary in 1788 and began his tenure as the U.S. president the same year, these roles came after the period in question.
> 5. Synthesizing information from various sources to produce an accurate answer.
>
> From inspecting the temporal graph corresponding to this question (Figure 3), it is evident that relying solely on the temporal graph would be insufficient to address all of the types of reasoning detailed above. Moreover, the text may contain irrelevant or potentially confusing information, like George Washington's roles or activities during other periods. Understanding negations within the context at times can further complicate the reasoning process. Thus, a language model needs not only to find the right information in the text but also to distinguish and filter out irrelevant or potentially misleading content.
>
> We appreciate you raising this question and will update the final version of our manuscript to include this discussion.
>
> > Q3: This concept merits exploration beyond its current application in temporal reasoning, showcasing potential generality and utility in broader reasoning contexts. Given its effectiveness in temporal reasoning, one may speculate about the feasibility of applying the same approach in other logical reasoning domains, such as spatial reasoning.
>
> Thank you for recognizing the generalizability of our method and for your insightful suggestions. While our current work focuses on temporal reasoning, we also see great potential in applying our proposed method to spatial reasoning. For instance, integrating it with geospatial information extraction systems to construct graph structures and fusing them into the input. We will expand our conclusion in the final version of our manuscript to include a discussion of such opportunities for future work.

---

### Official Review · Reviewer_D8zG · 2023-08-05

**Soundness:** 3

**Excitement:**

4: Strong: This paper deepens the understanding of some phenomenon or lowers the barriers to an existing research direction.

**Missing References:**

Aditya Sharma, Apoorv Saxena, Chitrank Gupta, Mehran Kazemi, Partha Talukdar, and Soumen Chakrabarti. "TwiRGCN: Temporally Weighted Graph Convolution for Question Answering over Temporal Knowledge Graphs." In Proceedings of the 17th Conference of the European Chapter of the Association for Computational Linguistics, pp. 2041-2052. 2023.

**Paper Topic And Main Contributions:**

This work focuses on the problem of time-sensitive question answering over the events and times in a question and an accompanying long context document. The authors use existing temporal information extraction systems to construct temporal graphs and investigate different fusion methods to inject them into transformer-based language models. They compare their input fusion method with prior approaches that use graph convolutions. Models are evaluated on SituatedQA and TimeQA datasets.

**Questions For The Authors:**

- Could you give some more qualitative examples like in Table 3?

**Reasons To Accept:**

- The paper is well-organised and easy to follow.
- Evaluation is interesting and beats state-of-the-art. In particular, Table 3 helps us understand the model behaviour qualitatively, though I wish there were more examples.
- The ChatGPT analysis is very intriguing.

**Reasons To Reject:**

- There are a lot of missing fields in Table 2, and most baselines have not been directly compared across datasets, so it’s difficult to generalise improvements.
- It can be argued that this is incremental work given model simplicity.

**Reproducibility:**

4: Could mostly reproduce the results, but there may be some variation because of sample variance or minor variations in their interpretation of the protocol or method.

**Reviewer Confidence:**

4: Quite sure. I tried to check the important points carefully. It's unlikely, though conceivable, that I missed something that should affect my ratings.

---

> ### Author Rebuttal · Authors · 2023-08-29
>
> Dear reviewer D8zG,
>
> Thank you for your review and the positive remarks on the organization of our paper, the intriguing aspects of our ChatGPT analysis, our evaluation setting, and the state-of-the-art performance of our method. Your mention of the qualitative insights from Table 3 is particularly appreciated. We address the insightful comments and questions you raised in your reviews below.
>
> > There are a lot of missing fields in Table 2, and most baselines have not been directly compared across datasets, so it’s difficult to generalise improvements.
>
> We appreciate your observation regarding missing values in Table 2. These missing values are associated with results that were directly reported in prior work. In cases where results are missing, this indicates that the prior work did not test on all of the datasets used in our experiments. Hence, values corresponding to those methods and datasets are not available.
>
> Nevertheless, for every dataset used in our experiments, we compared our results to all prior comparable work reported on that particular dataset. Our results indicate that our approach consistently outperforms the prior state-of-the-art models in various scenarios. This not only demonstrates that we tested on more datasets compared to prior works, but also further validates that our method can generalize across multiple datasets.
>
> Since our initial paper submission, we also replicated the FiD model on the SituatedQA dataset for further comparison. It has an exact match score of 23.7, which indicates that our LongT5_ERR method still significantly outperformed it. The experimental details are the same as the Replicated FiD baseline described in the paper, and we will add this new result to Table 2 in the final version of our manuscript. We will also update our discussion of Table 2 to clarify that missing values are associated with a lack of results reported in previous works.
>
> > It can be argued that this is incremental work given model simplicity.
>
> Thank you for raising this concern. We believe that the simplicity, intuitiveness, and effectiveness of the final graph fusion step is a strong advantage of ERR relative to more complicated fusion methods (e.g., GNN). Indeed, EMNLP follows the [ACL policy](https://2023.aclweb.org/blog/review-acl23/) in recognizing that "the goal is to solve the problem, not to solve it in a complex way" and that "simpler solutions are in fact preferable, as they are less brittle and easier to deploy in real-world settings."
>
> Our findings indicate that the most effective method for improving the temporal reasoning capabilities of transformers via temporal graphs is to incorporate them directly into the model's input. This can be achieved without the need for complex graph modeling techniques, such as those in previous state-of-the-art methods (e.g., DocTime). The effectiveness and simplicity of our approach provide valuable insights to the research community and can inform future work on graph fusion methods.
>
> > Could you give some more qualitative examples like in Table 3?
>
> Yes, thank you for this suggestion. Due to page constraints, we only displayed 5 out of our 50 manually annotated errors in Table 3 to illustrate each type of mistake. However, we will include more qualitative examples in the final version of our paper, and have provided some of the new examples in the table below. Model predictions and correct answers are in bold.
>
> | Type                                                       | Question                                                     | Relevant Context                                             | Prediction                                | Answer                                                |
> | ---------------------------------------------------------- | ------------------------------------------------------------ | ------------------------------------------------------------ | ----------------------------------------- | ----------------------------------------------------- |
> | False Negative                                             | Where was Richard G. Hovannisian educated from 1954 to 1958? | Hovannisian received his B.A . in history (1954) from the **University of California, Berkeley**, and his M.A . in history (1958) and his Ph.D . (1966) from **University of California, Los Angeles (UCLA).** | University of California, Berkeley        | University of California , Los Angeles (UCLA)         |
> | False Negative                                             | Which team did the player Simone Verdi belong to from 2014 to 2015? | On 18 June 2014, the co-ownership between Torino and **Milan** was renewed for a third year, with Verdi loaned to **Empoli** again. | Empoli                                    | Milan                                                 |
> | False Negative                                             | Segenet Kelemu was an employee for whom from Nov 2013 to Nov 2014? | In 2013, Kelemu joined the **Alliance for a Green Revolution in Africa** (AGRA) as Vice President for Programs for about a year. In November 2013, Kelemu became the Director General of the **International Centre of Insect Physiology and Ecology** (icipe) ... | Alliance for a Green Revolution in Africa | International Centre of Insect Physiology and Ecology |
> | Semantic Understanding                                     | Donald Thobega played for which team in 2006?                | ... Between 1999 and 2006 he won a total of five caps for the **Botswana national football team**. | N/A                                       | Botswana national football team                       |
> | Semantic Understanding                                     | Which school did Alexander Dvorkin go to from 1975 to 1978?  | ... in 1972, he became a student in the Faculty of Russian Language and Literature of **Moscow Pedagogical Institute** ... consequently, in the autumn of 1975, Dvorkin was expelled from the third year of the institute ... was expelled from the institute for poor academic performance and non-attendance in early 1975. | Moscow Pedagogical Institute              | N/A                                                   |
> | Semantic Understanding & Insufficient Temporal Information | What was the capital of Kolomna from 1929 to Apr 2017?       | Within the framework of administrative divisions, Kolomna serves as the administrative center of **Kolomensky District** ... | N/A                                       | Kolomensky District                                   |
> | Semantic Understanding & Insufficient Temporal Information | Where did Ryan Gander work from 2007 to 2015?                | ... In 2004, he was made Cocheme Fellow at Byam Shaw School of Art, **London** ... | N/A                                       | London                                                |
>
> > Missing References: Aditya Sharma, Apoorv Saxena, Chitrank Gupta, Mehran Kazemi, Partha Talukdar, and Soumen Chakrabarti. "TwiRGCN: Temporally Weighted Graph Convolution for Question Answering over Temporal Knowledge Graphs." In Proceedings of the 17th Conference of the European Chapter of the Association for Computational Linguistics, pp. 2041-2052. 2023.
>
> We appreciate this reference to interesting prior work, which we will cite in the final version of our paper. While Sharma's paper focuses on a different task (question answering over structured knowledge graphs vs. our focus on time-sensitive question answering over unstructured text), we will add the below discussion of how our work relates to it in Section 2.
>
> "Sharma et al. (2023) use pre-trained temporal knowledge graph embeddings combined with graph neural networks to reason over structured temporal knowledge graphs, with the goal of addressing temporal questions. In contrast, we concentrate on temporal reasoning in the reading comprehension setting, using unstructured text to deduce answers. As our method does not utilize a predefined structured knowledge base, our answers are derived from segments within unstructured texts rather than entities or relations in knowledge graphs."

---

### Official Review · Reviewer_XQnr · 2023-08-06

**Soundness:** 3

**Excitement:**

3: Ambivalent: It has merits (e.g., it reports state-of-the-art results, the idea is nice), but there are key weaknesses (e.g., it describes incremental work), and it can significantly benefit from another round of revision. However, I won't object to accepting it if my co-reviewers champion it.

**Paper Topic And Main Contributions:**

This paper designs a method to answer the time-sensitive questions. There are two key steps in this method. The first one is to use temporal extraction system to construct the temporal graphs. The purpose is to extract the event, time and temporal relations in advance and to construct a graph. With this structured temporal information, the second step is to investigate the way of fusing the temporal graph with the transformer models. The whole idea is clear and well explained in the paper.

**Questions For The Authors:**

Please check the three questions in the weaknesses.

**Reasons To Accept:**

Strengths:

1.	This paper proposes a way to build a temporal graph based on the temporal extraction method including CAEVO and SUTime. It will extract the importance temporal information from the text. This temporal graph can be a useful knowledge in many tasks.

2.	Two fusing methods are proposed in the paper. And they can be used easily with the large language models. The ERR approach outperforms the SOTA method.

**Reasons To Reject:**

Weaknesses:

1.	The idea to build the temporal graph is attractive in this paper. But the tech contribution of the whole paper is limited. The graph construction is based on the existing systems. And the fusing methods have been used in many tasks. These two fusing methods have some problems. First, The graph fusion has a bad performance as shown in Table 2. It is not a good candidate to take advantage of the structured temporal graphs. Second, the ERR is similar to a data processing step using the temporal graph. Since there is a step to extract the temporal information using other system, it is not supervised that the LongT5 with ERR can get a better performance than the LongT5.

2.	The EER method has one limitation. The cost of marking the nodes and edges of temporal graph is high if the input context is long. I am worried it is hard to be extended in large-scale data.

3.	The explanation of section 3.2.2 is not clear. What I got is that this method is coming from the Schlichtkrull’s paper. I am not clear if authors propose any modification so that it is more useful for temporal QA task.

**Reproducibility:**

3: Could reproduce the results with some difficulty. The settings of parameters are underspecified or subjectively determined; the training/evaluation data are not widely available.

**Reviewer Confidence:**

4: Quite sure. I tried to check the important points carefully. It's unlikely, though conceivable, that I missed something that should affect my ratings.

---

> ### Author Rebuttal · Authors · 2023-08-29
>
> Dear Reviewer XQnr,
>
> Thank you for your feedback and recognition of our paper's clarity, our temporal graph's generalization ability, our fusion method's compatibility with large language models, and the state-of-the-art performance of our ERR method. We appreciate your insightful comments and have addressed them below.
>
> > The idea to build the temporal graph is attractive in this paper. But the tech contribution of the whole paper is limited. The graph construction is based on the existing systems.
>
> Thank you for this feedback. While it is true that our temporal graph construction leverages existing systems, our goal is not to propose new temporal information extraction methods that can be used to construct graphs. Rather, our focus is on introducing an effective approach to construct and fuse graphs for a specific task: time-sensitive question answering, using the existing temporal information extraction systems.
>
> The novelty of our work is in its (1) investigation of whether fusion of temporal information extraction results can help neural time-sensitive question answering, and (2) investigation of which form of fusion yields the greatest benefits. To the best of our knowledge, there is no prior work investigating similar research questions. Our experiments surprisingly show that a simple but novel form of fusing temporal graphs into the input yields benefits where the popular technique of graph neural networks (GNNs) does not, which provides valuable insights to inform future work.
>
> > the fusing methods have been used in many tasks.
>
> In this work, we investigated two temporal graph fusion methods, namely ERR and GNN. Our ERR method is a novel and efficient temporal graph fusion approach that has not been proposed previously. It effectively enhances the model's temporal reasoning capabilities and consistently outperforms more complex methods, achieving state-of-the-art (SOTA) performance in multiple scenarios.
>
> Regarding the GNN method, we agree that it has been employed previously to fuse graphs for other tasks. This is actually why we sought to implement it in this work; specifically, our motivation was to use it as a baseline method for comparisons with our newly proposed ERR method. While GNNs have been successfully applied to other tasks, our results are interesting in that they show GNNs are not a good fit for time-sensitive question answering. The fact that ERR outperforms GNNs in our experiments suggests that the investigation of ERR as an alternative to GNNs in other tasks could be a promising direction for future work.
>
> > These two fusing methods have some problems. First, The graph fusion has a bad performance as shown in Table 2. It is not a good candidate to take advantage of the structured temporal graphs.
>
> Thank you for this accurate observation regarding the performance of the GNN method. As mentioned previously, our primary objective for the GNN method was to use it as an important baseline for comparison with our ERR method, given its widespread application and its natural fit for graph structure fusion. We agree with your assessment that the GNN method has bad performance, which we highlight in our paper and use as a basis for concluding that ERR is a more efficient method for temporal graph fusion  (lines 415 - 430 and lines 586 - 590).
>
> > Second, the ERR is similar to a data processing step using the temporal graph. Since there is a step to extract the temporal information using other system, it is not supervised that the LongT5 with ERR can get a better performance than the LongT5.
>
> We appreciate your feedback on our ERR method but believe that characterizing it as a data processing step oversimplifies the complexities of temporal graph fusion, which we address in our work. These complexities include the investigation of how to best use temporal information extraction systems to construct high-quality temporal graphs for time-sensitive questions and which parts of the temporal graph should be fused (please see lines 389 -398 and Appendix A.2 for details).
>
> Furthermore, the simplicity, intuitiveness, and effectiveness of the final graph fusion step are strong advantages of ERR relative to more complicated fusion methods (e.g., GNN). Indeed, EMNLP follows [ACL policy](https://2023.aclweb.org/blog/review-acl23/) in recognizing that "the goal is to solve the problem, not to solve it in a complex way" and that "simpler solutions are in fact preferable, as they are less brittle and easier to deploy in real-world settings."
>
> Our findings indicate that the most effective method for improving the temporal reasoning capabilities of transformers via temporal graphs is to incorporate them directly into the model's input. This can be achieved without the need for complex graph modeling techniques, such as those in previous state-of-the-art methods (e.g., DocTime). The effectiveness and simplicity of our approach provide valuable insights to the research community and can inform future work on graph fusion methods.
>
> > The EER method has one limitation. The cost of marking the nodes and edges of temporal graph is high if the input context is long. I am worried it is hard to be extended in large-scale data.
>
> Thank you for raising this concern about the scalability of our ERR method. As noted in Section 4.3 (lines 389-398) and Appendix A.2, we only need to fuse the document-time-to-question-time sub-graph of the temporal graph. Therefore, in the main performance tables of our paper, we reported the performance of models that fused this sub-graph. To clarify the input length before and after ERR fusion, we will add the following table to the Appendix of our paper, which shows that using ERR does not significantly increase the length of the input and, therefore, the computational cost.
>
> | Datasets                                       | Avg. # tokens before ERR Fusion | Avg. # tokens after ERR Fusion |
> |------------------------------------------------|------------------------------|-----------------------------|
> | TimeQA (combined all the modes and splits)     | 2617                         | 2862                        |
> | SituatedQA                                     | 9923                         | 11515                       |
>
> Additionally, the statistical details of all the datasets we evaluated are reported in Table 9 of Appendix A.6 and show that our experiments were conducted using large-scale long-document datasets. For instance, TimeQA Hard comprises over 14,000 training samples and around 3,000 test samples, with an average document length surpassing 2,500 tokens. Similarly, SituatedQA includes 6,009 training samples and approximately 3,000 test samples, with an average document length of around 10,000 tokens.
>
> > The explanation of section 3.2.2 is not clear. What I got is that this method is coming from the Schlichtkrull’s paper. I am not clear if authors propose any modification so that it is more useful for temporal QA task.
>
> The purpose of Section 3.2.2 is to provide a brief introduction to the GNN model, "relation graph convolution", that we used, and to explain how we employed it to fuse the temporal graph. We have made **no modifications** to the structure of this model. We appreciate the reviewer's suggestion and, if our paper is accepted, we will clarify this point in the revised version and consider renaming this section. Additionally, we will include more details about this GNN model in the appendix.

---

### Meta-Review · Area_Chair_7ua1 · 2023-09-15

**Recommendation:** 4

**Metareview:**

This work presents a method for addressing time-sensitive questions. The methodology comprises two steps: 1) a temporal extraction system of time and temporal relationships to construct a graph, and 2) the fusion of these temporal graphs with transformer models. Two fusion techniques are explored for this integration: directly as text, and using graph convolutions. The method is evaluated in SituatedQA and TimeQA datasets. The reviewers highlighted the strong empirical results, that the manuscript is well-organised and easy to follow. They also raised issues the work being incremental in relation to similar ideas and that some baseline evaluations were missing. During the rebuttal period those and other concerns were partially addressed by the authors who re-implemented an FiD model.

---

### Decision · Program_Chairs · 2023-10-07

**Decision:**

Accept-Findings

**Comment:**

This work presents a method for addressing time-sensitive questions. The methodology comprises two steps: 1) a temporal extraction system of time and temporal relationships to construct a graph, and 2) the fusion of these temporal graphs with transformer models. Two fusion techniques are explored for this integration: directly as text, and using graph convolutions. The method is evaluated in SituatedQA and TimeQA datasets. The reviewers highlighted the strong empirical results, that the manuscript is well-organised and easy to follow. They also raised issues the work being incremental in relation to similar ideas and that some baseline evaluations were missing. During the rebuttal period those and other concerns were partially addressed by the authors who re-implemented an FiD model.